# The *N*-Glycosylation of Total Plasma Proteins and IgG in Atrial Fibrillation

**DOI:** 10.3390/biom13040605

**Published:** 2023-03-28

**Authors:** Branimir Plavša, Janko Szavits-Nossan, Aleksandar Blivajs, Borna Rapčan, Barbara Radovani, Igor Šesto, Krešimir Štambuk, Vito Mustapić, Lovorka Đerek, Diana Rudan, Gordan Lauc, Ivan Gudelj

**Affiliations:** 1Faculty of Pharmacy and Biochemistry, University of Zagreb, 10000 Zagreb, Croatia; 2Magdalena Clinic for Cardiovascular Disease, Krapinske Toplice, Faculty of Medicine, J.J. Strossmayer University in Osijek, 31000 Osijek, Croatia; 3Faculty of Dental Medicine and Health, J.J. Strossmayer University in Osijek, 31000 Osijek, Croatia; 4Department of Cardiology, University Hospital Dubrava, 10000 Zagreb, Croatia; 5Department of Biotechnology, University of Rijeka, 51000 Rijeka, Croatia; 6Faculty of Medicine, J.J. Strossmayer University of Osijek, 31000 Osijek, Croatia; 7Clinical Department for Laboratory Diagnostics, University Hospital Dubrava, 10000 Zagreb, Croatia; 8Genos Glycoscience Research Laboratory, 10000 Zagreb, Croatia

**Keywords:** atrial fibrillation, *N*-glycosylation, immunoglobulin G, biomarker, total plasma proteins, pulmonary vein isolation

## Abstract

Atrial fibrillation is a disease with a complex pathophysiology, whose occurrence and persistence are caused not only by aberrant electrical signaling in the heart, but by the development of a susceptible heart substrate. These changes, such as the accumulation of adipose tissue and interstitial fibrosis, are characterized by the presence of inflammation. *N*-glycans have shown great promise as biomarkers in different diseases, specifically those involving inflammatory changes. To assess the changes in the *N*-glycosylation of the plasma proteins and IgG in atrial fibrillation, we analyzed the *N*-glycosylation of 172 patients with atrial fibrillation, before and six months after a pulmonary vein isolation procedure, with 54 cardiovascularly healthy controls. An analysis was performed using ultra-high-performance liquid chromatography. We found one oligomannose *N*-glycan structure from the plasma *N*-glycome and six IgG *N*-glycans, mainly revolving around the presence of bisecting *N*-acetylglucosamine, that were significantly different between the case and control groups. In addition, four plasma *N*-glycans, mostly oligomannose structures and a derived trait that was related to them, were found to be different in the patients who experienced an atrial fibrillation recurrence during the six-month follow-up. IgG *N*-glycosylation was extensively associated with the CHA_2_DS_2_-VASc score, confirming its previously reported associations with the conditions that make up the score. This is the first study looking at the *N*-glycosylation patterns in atrial fibrillation and warrants further investigation into the prospect of glycans as biomarkers for atrial fibrillation.

## 1. Introduction

Atrial fibrillation (AF) is a complex cardiac arrhythmia that is characterized by the irregular and rapid contraction of the atria, which results in a disorganized and ineffective cardiac output. Its prevalence has been steadily increasing in Western countries, even after adjusting for age, with this trend expected to continue in the future. Recent estimates indicate that 6% of individuals over the age of 65 and 10% of individuals over the age of 80 suffer from AF [1,2]. Furthermore, it has been estimated that approximately one third of patients that have been diagnosed with AF experience few or no symptoms, but still have a significantly higher risk of stroke and other AF-related complications [3]. AF has been identified as a major contributor to the global burden of cardiovascular disease, and it is associated with a significant morbidity, including stroke, heart failure, and a reduced quality of life [4]. Epidemiological studies have revealed several independent risk factors for the development of AF. These include age, hypertension, congestive heart failure, coronary artery disease, diabetes, left ventricular hypertrophy, and the male sex [5,6]. These conditions influence the disease’s onset and progression, while managing them reduces the risk for AF and improves patient outcomes [7].

The pathogenesis of AF is complex, multifaceted, and involves both electrophysiological changes in the heart, as well as structural and mechanical changes in the heart muscle itself. The initiating triggers for AF episodes mainly come in the form of the rapidly firing ectopic foci that are located in the muscular sleeves, which extend from the left atrium into the proximal parts of the pulmonary veins [8]. A pulmonary vein isolation procedure, therefore, is the cornerstone of treatment for patients with AF, but the recurrence rate remains high in patients with persistent AF and underlying structural changes of the left atrium. The structural changes of the heart include atrial dilatation, fibrosis, and the accumulation of epicardial adipose tissue. These changes form the substrate that is necessary for AF development and persistence [4]. Atrial fibrosis is the most prominent factor that is involved in the structural remodeling of AF. The fibrotic tissue that accumulates in the atria is stiff and poorly conductive, leading to the electrical and mechanical dysfunction of the atria [9]. Studies have shown that the development of atrial fibrosis is mediated by several mechanisms, such as inflammation, oxidative stress, and neurohormonal activation. Local inflammation leads to the activation of fibroblasts, which, in turn, produce collagen and extracellular matrix, resulting in an increase in fibrotic tissue [10]. The involvement of systemic inflammation is also a topic of great interest in the study of AF pathophysiology. Thus, circulatory inflammatory markers, such as C-reactive protein (CRP) and IL-6, have been found to be increased in AF patients [11]. These molecules have even been proposed as biomarkers for AF onset, as well as for AF recurrence after cardioversion or catheter ablation [12].

As inflammation plays a key role in the pathogenesis of AF, its understanding is important to investigate the molecular mechanism that underlies AF onset and progression. Inflammation is an intricately regulated process, with interactions of many different types of immune cells, signaling molecules, complement components, and circulating antibodies. These interactions are, in part, dependent on the *N*-glycosylation of the involved molecules [13]. *N*-glycosylation constitutes the co- and posttranslational addition of complex oligosaccharide structures (glycans) to proteins. It is estimated that more than half of all proteins are glycosylated [14]. Glycans influence the protein structure, function, and processing, and are crucial for many biological processes, including cell signaling and recognition [15,16]. It was shown that the total plasma protein and immunoglobulin G (IgG) *N*-glycosylation changes rapidly during acute inflammation [17]. Alterations in total the plasma protein and IgG glycosylation have also been reported in many other diseases and conditions, and have shown potential as biomarkers [18]. IgG in particular has been implicated in a variety of inflammatory and metabolic conditions, such as rheumatoid arthritis, obesity, cardiovascular disease (CVD), and aging [19,20]. IgG is an important modulator of the immune system, and through its effector functions, can have pro- or anti-inflammatory effects. In vitro studies have found that the differential N-glycosylation of the fragment crystallizable (Fc) region of IgG modifies the affinity of the IgG molecule to the Fc-receptors, modulating its pro- or anti-inflammatory function [21,22]. There is a growing body of research on the *N*-glycosylation changes in different cardiometabolic diseases. It was found that GlycA, a glycan-based plasma biomarker, correlates with the incidence of cardiovascular events [23], and that the same association has been observed for the total plasma protein and IgG *N*-glycome [24,25]. Besides cardiovascular events, IgG *N*-glycosylation has also been found to associate with the atherosclerotic risk score, as well as with the development of femoral and carotid plaques [24,26]. IgG glycosylation changes were also reported in hypertension [27]. In addition to different CVDs, *N*-glycans have been studied in diabetes, with both plasma and IgG *N*-glycans showing extensive alterations connected to the disease onset, as well as the progression of both type 1 and type 2 diabetes [24,28,29,30].

Given the aforementioned interplay between glycosylation and different CVDs, together with their risk factors, we conducted this study in order to explore, for the first time, the glycosylation alterations of the total plasma proteins, and additionally of isolated IgG, in AF, thus deepening our knowledge of the disease’s pathophysiology. Therefore, we recruited 172 participants with AF who underwent a catheter ablation and compared their total plasma protein and IgG *N*-glycosylation profiles with matching profiles from 54 cardiovascularly healthy controls, as well as with corresponding profiles six months after the ablation.

## 2. Materials and Methods

### 2.1. Subjects

A total of 226 individuals were included in this study, 172 patients with paroxysmal or persistent atrial fibrillation, and 54 healthy controls. The plasma samples from the atrial fibrillation patients were collected at the Magdalena Clinic, Krapinske Toplice, as described previously [31]. Briefly, the patients with AF who were indicated for a pulmonary vein isolation procedure using radiofrequency catheter ablation were recruited for the study, based on the following inclusion criteria: (1) 2 or more symptomatic episodes of atrial fibrillation, lasting for more than 5 min documented on an ECG; (2) age > 18; (3) an indication for a catheter ablation; and (4) informed consent provided. Patients with the following characteristics were not considered for inclusion: (1) permanent atrial fibrillation; (2) a previous catheter ablation; (3) a left ventricle ejection fraction of <50%, measured by an echocardiogram; (4) myocardial infarction; (5) primary cardiomyopathy; (6) an acute inflammatory condition, (7) CRP > 10 mg/L; (8) a chronic inflammatory condition; (9) a history of malignancy; and (10) hyper- or hypothyroidism. The patients were followed for AF recurrence for 6 months, using a transtelephonic ECG device. At the six-month follow-up clinical exam, a 24 h ECG holter monitoring was performed. The plasma samples were collected at two timepoints: before the procedure and at the six-month follow-up. The 54 plasma samples from the healthy controls were collected at the Dubrava Clinical Hospital, Zagreb. For all the participants, their venous blood samples were collected in vacuum blood collection tubes containing tri-potassium ethylenediaminetetraacetic acid (K3EDTA). The samples were allowed to rest for an hour and were then centrifuged at 1620× *g* for 10 min. The aliquots of the plasma were then transferred to a 2 mL tube, centrifuged at 2700× *g* for 10 min and immediately stored at −20 °C until the analyses were performed. All participants signed informed consent forms and the ethical committees of both institutions approved the research. The study was conducted in accordance with the Declaration of Helsinki.

### 2.2. Isolation of IgG from Human Plasma

Prior to the IgG isolation, all the samples were randomized across 96-well plates and 10 μL of the plasma was aliquoted for the total plasma protein analysis. The IgG was isolated from the plasma using an automated affinity chromatography method. This automation was performed using the Tecan Evo Freedom 200 instrument, coupled with a Resolvex A200 positive pressure unit. The IgG isolation method was based on a previously described manual method [32]. Briefly, 100 μL of the plasma was first filtered through a 96-well 0.45 mm SUPOR plate (Pall Corporation, New York, NY, USA), then diluted with 700 μL 1× phosphate-buffered saline (1×PBS, prepared in-house), and loaded onto the 96-well monolithic Protein G plate (Sartorius, Göttingen, Germany). The columns were repeatedly washed with 2 mL 1×PBS, and the bound IgG was eluted with 1 mL 0.1 M formic acid (Sigma-Aldrich, St. Louis, MO, USA). The IgG eluate was neutralized by adding 170 μL of 1 M ammonium bicarbonate (Sigma-Aldrich). The isolated IgG was dried down in a vacuum centrifuge and stored at −20 °C until it was ready for the analysis of the released glycans.

### 2.3. Enzymatic Release and Fluorescent Labeling of N-Glycans from IgG and Total Plasma Proteins

A high-throughput analysis of the IgG and total plasma protein *N*-glycans was performed using the HILIC-UHPLC method, described in detail by Pučić et al. and Trbojević-Akmačić et al. [32,33]. The dry, isolated IgG was resuspended in 30 μL of 1.33% SDS (Sigma-Aldrich), and a denaturation was performed via incubation at 65 °C for 10 min. The plasma samples were denatured by adding 20 μL of 2% SDS and incubating at 65 °C for 10 min. The rest of the procedure was the same for the IgG and total plasma protein *N*-glycan analysis. After this denaturation, 10 μL of 4% Igepal-CA630 (Sigma-Aldrich) was added and shaken at room temperature for 15 min. The enzymatic release of the *N*-glycans was performed by adding 1.2 U of PNGaseF (Promega, Madison, WI, USA) and incubating overnight at 37 °C. The released glycans were fluorescently labeled with 2-aminobenzamide (2-AB) by reductive amination. The labeling mixture consisted of 0.48 mg 2-AB (Sigma-Aldrich) and 1,12 mg 2-picoline borane (Sigma-Aldrich), in 25 μL 30% (*v*/*v*) acetic acid (Merck, Darmstadt, Germany) and dimethyl sulfoxide (Sigma-Aldrich). The labeling reaction was performed at 65 °C for 2 h. The labeled glycans were isolated from the reaction mixture solid-phase extraction (SPE). After this labeling, 700 μL of acetonitrile (ACN, VWR International, Radnor, PA, USA) was added to the samples, and the mixture was transferred onto a 96-well SUPOR filter plate (Pall Corporation, New York, NY, USA). The solvent was removed by the application of a vacuum on a vacuum manifold. The samples were washed 5 times with 96% ACN. The labeled *N*-glycans were eluted with ultra-pure water and stored at −20 °C until the chromatography analysis. 

### 2.4. Hydrophilic Interaction Liquid Chromatography of Labeled N-Glycans

The fluorescently labeled *N*-glycans were separated with hydrophilic interaction chromatography (HILIC) on an Acquity UPLC H-class instrument (Waters, Milford, MA, USA). The excitation and emission wavelengths were set to 250 nm and 428 nm, respectively. The plasma *N*-glycans and IgG *N*-glycans were separated on a 150 mm and 100 mm Glycan BEH Amide column (Waters, USA), respectively. 100 mmol/L solution of ammonium formate in water, with a pH of 4.4, was used as solvent A, and ACN was used as solvent B. For the plasma analysis, a linear elution gradient of 30–47% of solvent A, at a 0.56 mL/min flow rate in a 25 min analytical run, was used. The IgG was analyzed using a linear gradient of 25–38% of solvent A, at 0.4 mL/min flow rate in a 29 min analytical run. The data were processed using an automatic processing method, after which, each chromatogram was manually corrected to maintain consistent intervals of integration between the samples. The chromatograms were separated into 24 peaks (IGP1-IGP24) for the IgG *N*-glycans and 39 peaks (GP1-GP39) for the plasma *N*-glycans. The glycan composition of each peak was confirmed in previous studies using LC-MS [32,34]. The sample chromatograms, with major glycan structures in each peak for the total plasma and IgG *N*-glycans, are given in Appendix A.

### 2.5. Data Analysis

All the statistical analyses and data visualization were performed using R software version 4.0.5. The relative amount of each glycan in a sample was calculated as ratio of the corresponding peak area and total integrated area. In addition to the directly measured glycan traits, several derived traits were calculated by combining the peaks with certain common structural characteristics, such as fucosylation, galactosylation, sialylation, and bisecting N-acetylglucosamine. The formulas and descriptions of these derived traits are given in Appendix A. All the glycan data were log-transformed prior to the analysis, in order to obtain a normal distribution and to allow the use of parametric tests. To minimize the technical variations due to the analysis in several batches, a batch correction was performed using the ComBat method, R package “sva”. Linear models were used throughout to test for significant associations. Log-transformed glycans were used as the continuous dependent variable, and the variable that was tested for as the independent variable. Age and sex were included in the model as covariates for all the analyses, due to their known effect on the composition of the *N*-glycome of both the plasma proteins and IgG [35,36]. The CHA_2_DS_2_-VASc score for the stroke risk was calculated for all the AF patients, as described by Lip et al. [37]. Due to the small number of people with scores greater than 4, we combined all those scores into a single category, labeled as “>4”. The CHA_2_DS_2_-VASc was treated as an ordered categorical variable from 0 to >4 and the association was tested using a linear regression, with age and sex as covariates, the same as all the other analyses. Due to the correlated nature of the relatively quantified *N*-glycan measurements, for the adjustment of the *p*-values, we used the method described by Li and Ji, which computes the number of independent tests for which to adjust the *p*-value for [38]. The significance threshold was held at 0.05.

## 3. Results

The total plasma protein and isolated IgG *N*-glycosylation were analyzed in the 332 samples that were collected from the 172 people with AF that were scheduled for a catheter ablation procedure and the 54 healthy controls. Of those 172 patients, for 106 of them, a six-month follow-up plasma sample was obtained. The population characteristics of the study cohort are given in Table 1. An analysis of the 2-AB labeled *N*-glycans was performed using UHPLC, as described in the materials and methods section. From the relatively quantified chromatographic peaks, 16 derived traits for the plasma and 6 derived traits for the IgG were calculated, representing the proportions of the glycans that had certain common structural features. The formulas that were used for the calculation of these derived traits are given in Appendix A.

### 3.1. Associations of Plasma and IgG N-Glycans with AF

We assessed the differences in the *N*-glycosylation of the total plasma proteins and IgG between the AF patients and healthy controls. A regression analysis showed one total plasma protein *N*-glycan structure and seven IgG *N*-glycan structures that were significantly associated with atrial fibrillation. More precisely, in the plasma *N*-glycome, GP19 (Man9) was more abundant in the patients compared to the controls. In addition, in the IgG *N*-glycomes, IGP5 (Man5), IGP9 (FA2[3]G1), IGP20 (structure not determined), and IGP21 (A2G2S2) were more abundant in the AF patients compared to the controls, while the opposite was observed for IGP6 (FA2B), IGP10 (FA2[6]BG1), and the bisecting structures. The statistically significant associations are shown in Figure 1, while the results of the statistical analyses are given in Table 2. The associations for all the measured total plasma protein and IgG *N*-glycans are given in Appendix A, and shown in Appendix A.

### 3.2. Associations of Total Plasma Protein N-Glycans with AF Recurrence after Catheter Ablation

We also investigated the differences in the glycosylation between the patients who experienced a recurrence of AF after the catheter ablation during the six-month follow up. Because we were interested in the possible prediction of AF recurrence based on the *N*-glycan traits, we used the *N*-glycan data from the timepoint immediately before the procedure. In addition to the age and sex corrected model, we built a full model, with diabetes and hypertension as additional covariates. This addition had a minimal effect on the first model. Both models are given in Appendix A. In the plasma *N*-glycome, we found five *N*-glycan structures and one *N*-glycosylation trait that were significantly increased in the patients with AF recurrence, compared to those who did not experience recurrence. More precisely, GP2 (Man5, FA2B), GP7 (Man6), GP9 (A2BG2), GP15 (A2BG2S1), and the total oligomannose structures were increased. GP6 (FA2[6]BG1) and the overall bisecting structures also showed an increase and were nominally significant; however, these findings were attenuated after the correction for multiple testing, with the adjusted *p*-values being 0.06 and 0.07, respectively. We found no significant association with the IgG *N*-glycans. The statistically significant glycan differences are shown in Figure 2A, while detailed results are given in Table 3. The associations for all the measured plasma *N*-glycans are given in Appendix A.

### 3.3. Differences in IgG N-Glycome following Catheter Ablation

We found no association between the post-procedure total plasma protein or the IgG *N*-glycans and AF recurrence. The change in the total plasma protein *N*-glycans, pre- and post-procedure, also showed no association with recurrence. We did, however, notice changes in the four IgG *N*-glycans between the pre- and post- catheter ablation sampling. Thus, IGP5, IGP17 (A2G2S1), IGP20, and IGP21 decreased in abundance following the procedure. These changes are shown visually in Figure 2B, and the results are given in Table 3 for the significant *N*-glycans. The associations for all the measured IgG *N*-glycans are given in Appendix A.

### 3.4. Association of IgG N-Glycans with the CHA_2_DS_2_-VASc Score

For all the AF patients, we calculated the CHA_2_DS_2_-VASc score for the stroke risk [37]. We tested the association between the *N*-glycan traits and the CHA_2_DS_2_-VASc score using a linear regression. We found no significant association with the total plasma protein *N*-glycans, while the four IgG *N*-glycan traits and six *N*-glycan structures were significantly associated with the CHA_2_DS_2_-VASc score. Of the directly measured IgG *N*-glycans, IGP4 (FA2) and IGP6 (FA2B) showed an increasing linear trend, while IGP8 (FA2[6]G1), IGP14 (FA2G2), IGP18 (FA2G2S1), and IGP23 (FA2G2S2) went in the opposite direction. Of the IgG derived traits, the bisecting and agalactosylated structures showed an increasing trend, while the digalactosylation and sialylation were found to be decreasing. The results of this linear regression are given in Table 4, while the significant associations are shown in Figure 2C. The associations for all the measured IgG *N*-glycans are given in Appendix A.

## 4. Discussion

To our knowledge, this is the first study to investigate the *N*-glycosylation of the total plasma proteins and IgG in AF. Both the total plasma protein and IgG *N*-glycosylation exhibited changes in the patients with AF compared to the controls, but these changes were more extensive when looking at the IgG *N*-glycosylation. On the other hand, the total plasma protein *N*-glycosylation was significantly associated with AF recurrence after a catheter ablation.

For the plasma *N*-glycome, the alterations in AF were confined to a single glycan, Man9, which was found to be more abundant in the AF patients compared to the controls. A study by Cvetko et al. found the same structure to be increased in incident insulin resistance and type 2 diabetes when compared to healthy controls [30]. This could be due to a shared pathophysiological pathway between these two diseases, as diabetes is a known risk factor for AF [39]. Man9 is an oligomannose glycan, and in plasma, it is thought to come from apolipoprotein B-100 (ApoB-100) [40]. High-mannose structures, including Man9, have also been reported on the Fab portion of IgG and the C3 complement component, with the latter exclusively featuring oligomannose structures [41]. The presence of oligomannose glycans on vascular endothelial cells has been associated with inflammation, and plays a role in leukocyte trafficking [42]. Due to the relative quantification of glycans, it is not possible to distinguish if these changes come from alterations in the overall *N*-glycosylation or from an abundance of change in a specific protein. A study looking at the ApoB levels in AF found that patients with AF had lower overall levels of ApoB compared to healthy controls, thus pointing to the *N*-glycosylation change, as opposed to the protein levels [43]. This difference may also be due to C3 as a consequence of inflammatory changes. Further protein specific analysis is, however, required to elucidate the origin of the changes we have described herein.

More extensive changes were seen in the IgG *N*-glycome. We observed a higher abundance of Man5, FA2[3]G1, and A2G2S2 structures, as well as a lower abundance of FA2B, FA2[6]BG1, and bisecting *N*-glycan structures, in the AF patients compared to the controls. Many of these glycan structures were associated with different conditions, such as diabetes and CVD, that contribute AF pathology, yet these associations differed in several ways. The most unique change was the decrease in the bisecting structures, stemming mainly from the decrease in the FA2B and FA2[6]BG1. An increased bisecting is noticed throughout the literature, in studies looking at the IgG *N*-glycosylation in inflammation, diabetes, CVD, and hypertension, which are all states that are associated with AF [24,27,28]. Furthermore, higher levels of bisecting N-acetlyglucosamine have been associated with proinflammatory IgG changes due to an increased binding to FcγRIII, and consequently, an increased antibody-dependent cellular cytotoxicity (ADCC) [44]. The changes in FA2[3]G1 are also of the opposite direction compared to those that are noticed in diabetes and CVD [24,28,29]. Overall, an inflammatory IgG profile, present in these conditions and consisting of decreased galactosylation and sialylation, is not reveled in AF. The *N*-glycosylation changes in IgG most closely resemble type 1 diabetes, with a study by Rudman et al. also finding an increase in Man5, IGP20, and A2G2S2, with the already mentioned difference in the change of the bisecting glycans [28]. These similarities, noticed also in the plasma, cannot be merely due to the prevalence of diabetes in the AF group, as it is relatively low (8%) [45]. 

Several total plasma protein *N*-glycans were found to be increased in the patients that experienced a recurrence of AF after a catheter ablation. These *N*-glycan structures were FA2B, A2BG2, and A2BG2S1, which are glycans that contain a bisecting N-acetylglucosamine, and Man5 and Man6, which are oligomannose glycans. Overall, the oligomannose structures also showed an increase. In addition to the already mentioned significantly increased structures, a bisecting structure, FA2[6]BG1, and the overall bisecting structures were also increased, but were only nominally significant and did not reach statistical significance after correction, which could be due to the insufficient power of the study. Interestingly, an increase in plasma bisecting and high-mannose structures was found in the previously mentioned study by Rudman et al., comparing the *N*-glycosylation of children with type 1 diabetes to their healthy siblings, where the alterations of the same structures, FA2B, Man5, and Man6, were observed [28]. In our study, these associations remained significant even after correcting for the presence of diabetes or hypertension, as shown in Appendix A. Thus, this indicates the existence of a different pathophysiological process that contributes to those glycosylation changes. An increase in plasma bisecting structures is indicative of a pro-inflammatory *N*-glycan profile. Additionally, as previously discussed, oligomannose glycans have been implicated in diabetes and are present on the C3 complement component, which could also indicate an inflammatory state that predisposes patients to poor procedure outcomes. This is also consistent with previous research, which looked at CRP and other inflammatory biomarkers as predictors of AF recurrence, and found a positive association [46]. It is also noteworthy that we excluded the patients with acute inflammation and an increased hsCRP, but still observed differences that were consistent with an increased inflammatory state, namely increased bisecting glycans. This may indicate that plasma glycans are relatively sensitive biomarkers of inflammation and AF recurrence.

The changes that we observed six months following the catheter ablation consisted of a decrease in Man5, A2G2S1, IGP20 (structure not determined), and A2G2S2. Interestingly, in our study, Man5, IGP20, and A2G2S2 were also lower in the controls compared to the patients with AF. This may indicate a return to a “healthy” glycan profile. The level of change was similar in both those who experienced an AF recurrence and in those who did not. This could be due to the fact that, most patients, even those who experienced AF recurrence, had a relatively long period of being disease-free.

The CHA_2_DS_2_-VASc score is calculated based on the presence of several clinical risk factors for stroke, with these being: congestive heart failure, hypertension, age, diabetes mellitus, prior stroke or thromboembolism, vascular diseases such as aortic plaque or myocardial infarction, and the female sex. *N*-glycosylation changes were described in most of these conditions, and therefore it is unsurprising that the IgG *N*-glycosylation traits were extensively associated with the score. Our results show a compounding effect of different risk factors on the glycan composition; as more risk factors are present, there is a higher abundance of agalactosylated, asialylatied, and bisected *N*-glycan structures, such as FA2 and FA2B. Both of these structures are increasing and their increase was previously reported in cardiovascular disease risks [26]. The rest of the structures that showed significant associations, FA2[6]G1, FA2G2, FA2G2S1, and FA2G2S2, are all decreasing, and are either galactosylated or sialylated. It is thought that sialylation plays a key role in the switch from the anti- to pro-inflammatory effects of IgG, through the modulation of binding for the activating Fc receptors. A study by Kaneko et al. found that the sialylated glycoforms of mouse IgG have a decreased affinity for the activating receptors FcγRIII and FcγRIV [47]. Because terminal galactose is necessary for the addition of sialic acid to the glycan, decreased galactosylation results in a decrease in sialylation. Decreased galactosylation itself has been proposed to increase the binding to the mannose binding lectin, due to the greater exposure of the mannose residues that are present in the core structure of *N*-glycans. In rheumatoid arthritis, this was shown to increase the activation of the complement via the MBL pathway [48]. Increased IgG galactosylation also promotes cooperative FcγRIIIb signaling with dectin-1, suppressing the proinflammatory signaling of the C5aR and CXCR2 pathways [49]. The association of IgG *N*-glycans with the CHA_2_DS_2_-VASc score is perhaps unsurprising, but confirms once again the extensive changes in the IgG *N*-glycosylation of different diseases with an underlying inflammatory pathology.

While some of the changes in the *N*-glycosylation patterns of AF are common to the other diseases that are associated with AF, such as diabetes, overall, AF exhibits a unique *N*-glycan signature, where a decrease in bisecting structures and an increase in FA2[3]G1 show a departure from the changes that have been observed in atherosclerotic CVDs and diabetes. This could offer potential for the use of IgG *N*-glycans as tools for the early detection of AF based on *N*-glycan biomarkers, but larger studies would have to be conducted. The changes in the plasma *N*-glycans that are connected to AF recurrence after catheter ablation are noticeable, and could signify an increase in certain inflammatory proteins, or the inflammatory response in general. We consider this to be a tentative first step toward further research on the possibility of using *N*-glycans, perhaps in combination with other biomarkers, to predict AF recurrence after catheter ablation, and therefore to guide treatment options. Due to this being the first, and a relatively small study, its findings will have to be replicated in larger cohort studies to be able to construct a predictive glycan-based model. Lastly, while the changes in the *N*-glycosylation that are described herein have certain biological interpretations, these findings need to be further investigated in protein-specific analyses and functional studies. 

## Figures and Tables

**Figure 1 biomolecules-13-00605-f001:**
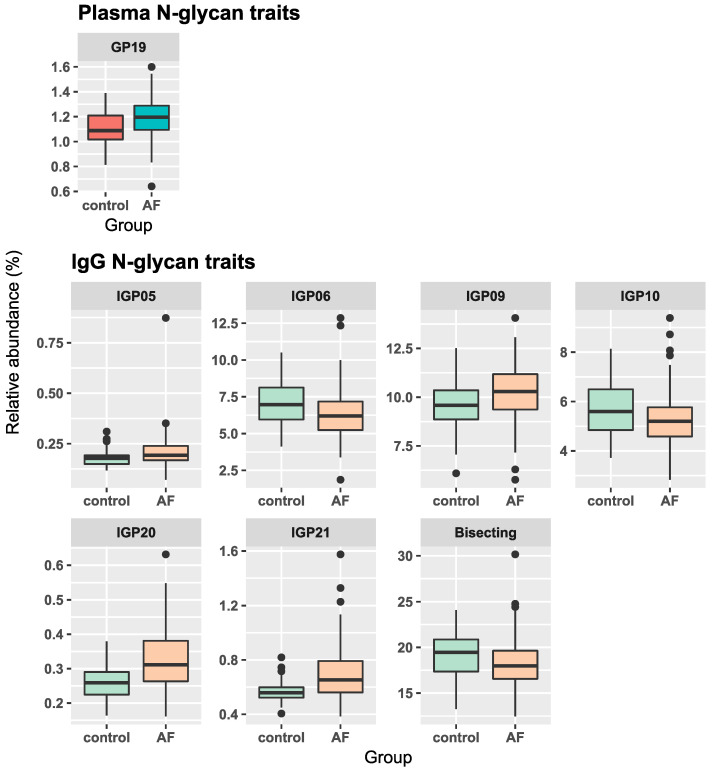
Statistically significant differences in *N*-glycan traits in plasma and IgG between the control group and AF group. Data are shown as boxplots, the line represents the median value, the box extends from the 25th to the 75th percentile, and the lines extend to the 1.5 × IQR. Outliers outside 1.5 IQR are plotted individually as dots.

**Figure 2 biomolecules-13-00605-f002:**
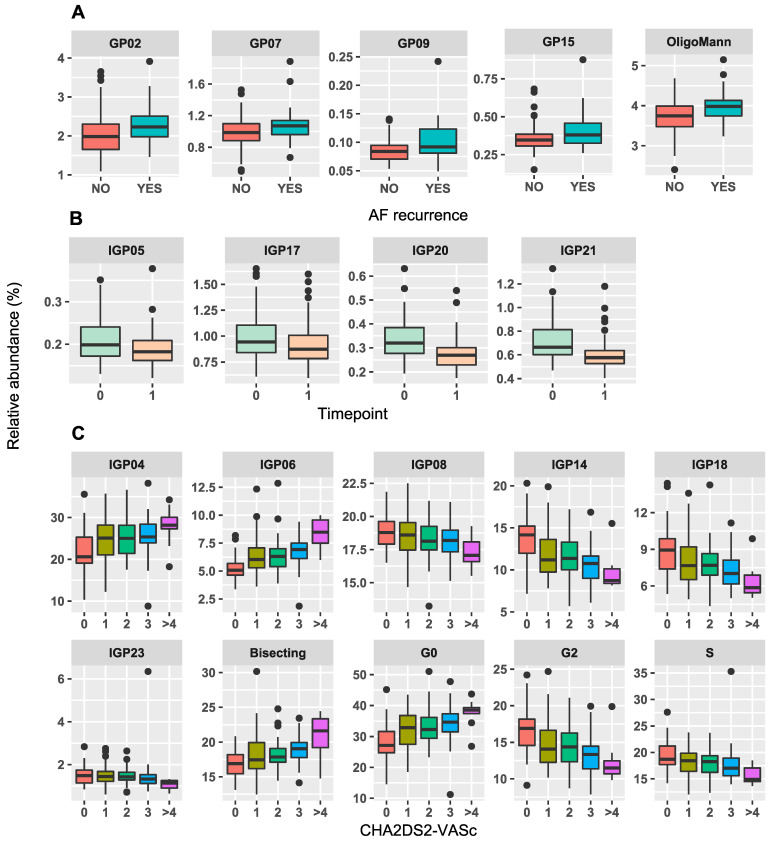
Association of total plasma protein *N*-glycosylation with AF recurrence (**A**). Association of IgG *N*-glycosylation between the pre- and post-procedure measurements (**B**). Association of IgG *N*-glycosylation with the CHA_2_DS_2_-VASc score (**C**). Data are shown as boxplots, as described in Figure 1. Only statistically significant structures and traits are shown.

**Table 1 biomolecules-13-00605-t001:** Study population characteristics.

		Atrial Fibrillation	Control
N		172	54
Age (years)		64 (57–69)	64 (57–69)
Female sex		63 (36%)	19 (35%)
BMI		28.9 (4.37)	
AF classification			
	Paroxysmal	135 (78%)	
	Persistent	37 (22%)	
CHA_2_DS_2_-VASc			
	0	33 (19%)	
	1	48 (28%)	
	2	45 (26%)	
	3	36 (21%)	
	<4	10 (6%)	
Hypertension			
	No	58 (34%)	
	Yes	114 (66%)	
Diabetes melitus			
	No	159 (92%)	
	Yes	13 (8%)	
Coronary disease			
	No	153 (89%)	
	Yes	19 (11%)	
Stroke or TIA			
	No	165 (96%)	
	Yes	7 (4%)	
AF recurrence (6 months)			
	No	120 (70%)	
	Yes	50 (29%)	
	N/A	2 (1%)	
Atrial fibrosis ^#^			
	No	21 (12%)	
	Mild	43 (25%)	
	Extensive	103 (60%)	
	N/A	5 (3%)	

Age is given as the median and interquartile range. All other categorical variables are presented as the N (%). BMI is presented as mean (SD). TIA—transient ishemic attack, AF—atrial fibrillation, and BMI—body mass index. ^#^ Presence of atrial fibrosis measured using electroanatomical voltage mapping, as described in [31]. Mild fibrosis is defined as percentage of fibrotic area in total left atrial surface area < 5%, while >5% is characterized as extensive.

**Table 2 biomolecules-13-00605-t002:** Differences of total plasma proteins and IgG *N*-glycosylation between patients with AF and healthy controls. Only statistically significant traits and *N*-glycan structures are shown in the table.

Sample	Glycan Trait	β-Coefficient (95% Confidence Interval) *	*p*-Value	Adjusted *p*-Value ^#^
Plasma	GP19	0.075 (0.036, 0.114)	0.0002	0.004
IgG	IGP5	0.123 (0.042, 0.204)	0.003	0.012
	IGP6	−0.093 (−0.167, −0.019)	0.0134	0.022
	IGP9	0.065 (0.02, 0.109)	0.0048	0.047
	IGP10	−0.083 (−0.143, −0.024)	0.0062	0.029
	IGP20	0.221 (0.149, 0.294)	8 × 10^−9^	1 × 10^−7^
	IGP21	0.166 (0.095, 0.237)	6 × 10^−6^	6 × 10^−5^
	Bisecting	−0.055 (−0.096, −0.014)	0.0084	0.037

* β-coefficient represents the natural logarithm of the relative change in *N*-glycan traits between groups corrected for age and sex differences. ^#^ *p*-value was adjusted using the Li-Ji correction method.

**Table 3 biomolecules-13-00605-t003:** Differences of total plasma proteins *N*-glycosylation between patients with recurrent AF and those without recurrence, and IgG *N*-glycosylation differences before and 6 months after catheter ablation. Only statistically significant traits and *N*-glycan structures are shown in the table.

Sample	Association	Glycan Trait	β-Coefficient (95% Confidence Interval) *	*p*-Value	Adjusted *p*-Value ^#^
Plasma	AF recurrence	GP2	0.117 (0.038, 0.196)	0.0041	0.034
		GP7	0.094 (0.028, 0.159)	0.0053	0.037
		GP9	0.123 (0.039, 0.207)	0.0043	0.034
		GP15	0.12 (0.046, 0.195)	0.0017	0.024
		OligoMan	0.069 (0.03, 0.107)	0.0006	0.012
IgG	Pre- and post-procedure	IGP05	−0.108 (−0.166, −0.05)	0.0003	0.002
		IGP17	−0.086 (−0.143, −0.029)	0.0031	0.018
		IGP20	−0.19 (−0.249, −0.13)	2 × 10^−9^	2 × 10^−8^
		IGP21	−0.172 (−0.227, −0.118)	2 × 10^−9^	2 × 10^−8^

* β-coefficient represents the natural logarithm of the relative change in *N*-glycan traits between groups corrected for age and sex differences. ^#^ *p*-value was adjusted using the Li-Ji correction method.

**Table 4 biomolecules-13-00605-t004:** Associations of IgG *N*-glycans with the CHA_2_DS_2_-VASc score corrected for age and sex. Only statistically significant traits and *N*-glycan structures are shown in the table.

Glycan Trait	Linear β-Coefficient (95% Confidence Interval)	*p*-Value	Adjusted *p*-Value ^#^
IGP4	0.179 (0.04, 0.319)	0.0118	0.034
IGP6	0.355 (0.201, 0.508)	1 × 10^−5^	1 × 10^−4^
IGP8	−0.067 (−0.119, −0.015)	0.0123	0.034
IGP14	−0.213 (−0.352, −0.074)	0.0028	0.014
IGP18	−0.186 (−0.323, −0.048)	0.0086	0.029
IGP23	−0.275 (−0.475, −0.076)	0.007	0.027
Bisecting	0.132 (0.049, 0.215)	0.0019	0.013
G0	0.214 (0.087, 0.341)	0.0011	0.011
G2	-0.184 (−0.315, −0.053)	0.006	0.026
S	-0.152 (−0.248, −0.055)	0.0023	0.013

^#^ *p*-value was adjusted using the Li-Ji correction method.

## Data Availability

The data presented in this study are available upon request from the corresponding authors.

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
