# Peer review of "The N-Glycosylation of Total Plasma Proteins and IgG in Atrial Fibrillation"

_biomolecules, 2023, doi:10.3390/biom13040605_

Round 1

Reviewer 1 Report

Glycosylation is one of the most common and important posttranslational modifications of proteins. This manuscript explores the use of HILIC-UHPLC as a high-throughput method for the identification of N-glycosylation in atrial fibrillation. The authors recruited 172 participants with atrial fibrillation who underwent catheter ablation and compared their total plasma protein and IgG N-glycosylation profile with matching profiles from 54 cardiovascularly healthy controls as well as with corresponding profiles six months after ablation. Overall, the authors did a nice job and this is an interesting paper. The data are in general of high quality; however, some questions need to be addressed.

1. HILIC-UHPLC has already been used by Radovani et al. Biomolecules 2023, 13, 375. The two articles use similar methods. Thus, the technology lacks sufficient innovation in this study.

2. The authors only confirm that there are differences in N-glycosylation. Further studies need to be conducted to determine the biological functions of N-glycans in atrial fibrillation.

Author Response

Thank you for your comments, please find our response below and also consider the changes we made to the manuscript as described in our response.

Point 1: HILIC-UHPLC has already been used by Radovani et al. Biomolecules 2023, 13, 375. The two articles use similar methods. Thus, the technology lacks sufficient innovation in this study.

Response 1: HILIC-UHPLC is the most common and robust method for the analysis of glycans from human plasma proteins and isolated human IgG. It was therefore the method of choice to generate an insight in the changes of glycosylation present in atrial fibrillation. This study uses the same method as Radovani et al. to answer a different question.

Point 2: The authors only confirm that there are differences in N-glycosylation. Further studies need to be conducted to determine the biological functions of N-glycans in atrial fibrillation.

Response 2: We agree with your comment, and we did mention this in our manuscript briefly in the discussion at the end of the second paragraph. This was the first study to look at whether N-glycosylation exhibits any changes at all in atrial fibrillation, and before any protein specific or functional studies we felt it was prudent to investigate whether any N-glycosylation changes exist in atrial fibrillation. Thus, we feel that reporting these findings is useful to inform further research into this topic. We emphasised it by adding the following to the concluding paragraph of the paper "Lastly, while the changes in N-glycosylation described herein have certain biological interpretations these findings need to be further investigated in protein specific analyses and functional studies.". We hope this makes it clearer that further research is necessary for biological interpretation of glycosylation changes.

We again thank you for your comments and hope that we have addressed your concerns adequately.

Reviewer 2 Report

Authors present  N-glycomic studies on total plasma and IgG in a group (172 patients) with atrial fibrillations before and six months after pulmonary vein isolation using HPLC. The experimentals were well done by a well known group operating in the field from many years. Results are well presented and the discussion well done. In my opinion the manuscript deserves publication in its actual form.

Author Response

We thank you for your kind comments.

Reviewer 3 Report

In this work, Branimir and colleagues report the N-glycan biomarkers of plasma proteins and IgG in atrial fibrillation with 172 patients and 54 healthy controls. The IgG was isolated from plasma and glycan was analyzed after derivatization. Over all the work is of high quality and well-presented. However, I do have some minor comments and recommendations I think the authors should consider prior to publication. 

1. On page 3, line 108, please delete one "with"

2. On page 7, line 255, please explain why the adjusted p-value thresholds are set to 0.06 and 0.07 respectively.

Author Response

We thank you for your comments and hope we addressed your comments appropriately.

Point 1: On page 3, line 108, please delete one "with".

Response 1: The error has been corrected in the revised manuscript.

Point 2: On page 7, line 255, please explain why the adjusted p-value thresholds are set to 0.06 and 0.07 respectively.

Response 2: The significance threshold was still at p < 0.05, we report that these glycans were significant before adjustment but were just outside the cut-off point after adjustment for multiple testing, hence we use the term nominally significant. We felt that it was valuable to report these findings as they are in accordance with other changes that were statistically significant even after correction. GP6 (nominally significant) and GP2, GP15, and GP15 (significant even after correction) all contain bisecting structures. The changes are in the same direction also, so the increase in the bisecting trait (nominally significant) can be expected. We also comment in the discussion that this could be due to the insufficient power of the study. We have adjusted the discussion paragraph to make it clearer that these findings specifically are only nominally significant (Page 11 lines 337-344 in the revised manuscript).

Round 2

Reviewer 1 Report

 None